# Surgical Skills and Technological Advancements to Avoid Complications in Lateral Neck Dissection for Differentiated Thyroid Cancer

**DOI:** 10.3390/cancers13143379

**Published:** 2021-07-06

**Authors:** Aldo Bove, Maira Farrukh, Adele Di Gioia, Velia Di Resta, Angelica Buffone, Claudia Melchionna, Paolo Panaccio

**Affiliations:** 1Department of Medicine, Dentistry and Biotechnology, University “G. D’Annunzio”, Via dei Vestini 31, 66100 Chieti, Italy; maira.farrukh@yahoo.it (M.F.); buffoneangelica89@gmail.com (A.B.); clamelchionna@gmail.com (C.M.); paolo.panaccio@gmail.com (P.P.); 2Unit of General Surgery, Pierangeli Hospital, 65124 Pescara, Italy; adeledigioia1@virgilio.it (A.D.G.); diresta.velia@gmail.com (V.D.R.)

**Keywords:** differentiated thyroid cancer, lateral neck dissection, nerves injuries, vascular injuries, surgical skills

## Abstract

**Simple Summary:**

The indication for performing a neck dissection in patients with differentiated thyroid cancers is metastatic involvement of the cervical nodes. Potential complications from lateral neck dissection include injury to the thoracic duct, cervical major vessels and nerves. There is a lack of review reporting the incidence of these complications and their diagnostic and therapeutic approach. Our review may help endocrine surgeons and clinicians to recognize and face up to these challenging problems.

**Abstract:**

Neck dissection is a surgical procedure reserved for thyroid cancer cases with clinically evident lymphatic invasion. Although neck dissection is a reliable and safe procedure, it can determine a significant morbidity involving a variety of structures of nervous, vascular and endocrine typology. A careful pre-operative study is therefore essential to better plan surgery. Surgical experience, combined with accurate surgical preparation and merged with adequate and specific techniques, can certainly help reduce the percentage of complications. In recent years, however, technology has also proved to be useful. Its crucial role was already recognized in the safeguard of the integrity of the laryngeal nerve through neuro-monitoring, but new technologies are emerging to help the preservation also of the parathyroid glands and other structures, such as the thoracic duct. These surgical skills combined with the latest technological advancements, that allow us to reduce the incidence of complications after neck dissection for thyroid cancer, will be reported in the present article. This topic is of significant interest for the endocrine and metabolic surgeons’ community.

## 1. Introduction

Papillary carcinoma seems to be the most frequent tumor originating from the thyroid gland [1] and, even if its prognosis is described as favorable [2], there are many problems associated with the surgical treatment of this type of neoplasms. Lymph node metastases are extremely widespread (20–60%) [3] and their presence does not seem to influence long-term prognosis [4]. They can be detected at the time of the first diagnosis of the neoplasms or in any subsequent follow-up of the patient. The role of the presence of these so called loco-regional metastases on the prognosis is still controversial, although it is now accepted by all the indication to intervene only in the presence of clinically palpable nodules [5]. In these cases, the surgeon must try to gain an effective regional control of the disease and avoid damaging the structures present in the neck. Anatomically, the lymph node stations are divided into six levels, including the pre-tracheal region. The initial attempts of the therapeutic removal of the affected lymph nodes date back to 1906 when radical latero-cervical lymphadenectomy was first described in 1906 by George Crile and implied not only the removal of lymph nodes from levels I–V but also the sacrifice of the deep jugular vein, the spinal accessory nerve and the sternocleidomastoid muscle [6]; later, a functional lymphadenectomy was described by the school of Osvaldo Suarez, that allowed the preservation of the above-mentioned structures and it is currently the approach used for differentiated thyroid cancer [7]. Despite the well-defined technique, the incidence of complications still remains high [8] and includes both the usual sequelae of thyroid surgery, such as hypoparathyroidism or injury to the lower laryngeal nerves, and also injuries to other structures in the region, such as the thoracic duct, the spinal accessory nerve or the carotid artery.

In order to lower the incidence of these complications, the extent of the lymphadenectomy was reduced, excluding portions II B and V A, allowing the risk, however, of a high incidence of recurrence [9,10]. An effective preoperative study can certainly help to better plan the surgical procedure and the surgeon’s experience is essential in limiting the incidence of complications. At the same time the development of technologies has produced significant benefits, one need only think of neuro-monitoring (NIM) or fluorescence.

## 2. Materials and Methods

Complications after neck lymphadenectomy for differentiated thyroid cancer can vary depending on the characteristics of the pathology (location, extension, invasiveness) and factors associated to the technique (re-operations, enlarged exeresis). Complications of neck dissection for differentiated thyroid cancers are reported in Table 1. Lymph node level and landmarks of the neck and are reported schematically in Figure 1. This paper provides a comprehensive and schematical review of nerve injuries, problems regarding the preservation of the parathyroid glands, thoracic duct injuries and the less frequent vascular injuries according to the most accurate study methods and the latest technological advancements developed in order to reduce the incidence of complications in lymphadenectomy (neck dissection) for differentiated thyroid cancer.

## 3. Results

### 3.1. Nerve Injuries

In neck dissection surgery, many nervous structures can be involved and eventually damaged (scheme) although the most frequent injuries concern the lower laryngeal nerve (temporary 7–8%, permanent 2–3%).

The surgical technique implies the identification and careful dissection of the nerve, but injury can also be caused indirectly by traction maneuvers or from the use of thermal energy devices. The intraoperative monitoring technique of the inferior laryngeal nerve is now practiced almost routinely and has been studied especially for benign thyroid surgery. Identification of the inferior laryngeal nerve is also possible.

Currently, meta-analyses do not highlight a clear advantage of intraoperative neurophysiological monitoring (“IONM”) in reducing the incidence of nerve damage during total thyroidectomy [11]. The continuous recording technique certainly seems to be more promising, especially in cases of re-operation [12]. During re-operations, it is also good practice to search for the nerve in the so-called “free segment”, that is, in the anatomical area not affected by the first operation. The incidence of non-recurrent inferior laryngeal nerve is low but can be prevented with a pre-operative CT study which can highlight an abnormal course of the right subclavian artery [13]. In this case, IONM seems to be very useful in identifying the nerve [14].

Spinal accessory nerve injuries vary according to the type of neck dissection performed (4.8–27%) and are more frequent in the case of dissection of levels II B and V A [15]. The pre-operative study, through ultrasounds or MRI, helps to identify the nerve and to study its course with regard to the surrounding structures [16].

The anatomical knowledge of the course of the nerve with its variables, especially regarding the relationship with the deep jugular vein, is necessary for an accurate and thorough surgical preparation. It is in fact known that, in its proximal portion, the course (30%) of the vein is set anterior to the nerve, thus facilitating damages to the nerve itself, while, when set posteriorly to the nerve (70%), its isolation and traction on loops is facilitated. Moreover, in 96% of the cases, the spinal accessory nerve is oriented, at the level of the upper edge of the posterior section of the digastric muscle, laterally with respect to the internal jugular vein, in 3% it is medial to it and in 1% it passes through the vein itself [17].

For the purpose of identifying the nerve, it is important to use IONM (with the contraction of the trapezius muscle), especially when treatment requires a dissection and clearing up of the level II B where lymph nodes lie above and behind the nerve [18].

In this case, it is necessary to recognize the different subdivision pathways of the spinal accessory nerve in relation to the sternocleidomastoid muscle: in type 1 (66%), the branches for the trapezius muscle are located at the posterior edge of the muscle; in type 2 (22%), they are present before the nerve enters the muscle; in type 3 (12%), the motor branches for the trapezius muscle that were formed on the posterior border of the sternocleidomastoid muscle advance medially and fuse with the cervical plexus [19].

The use of IONM allows the identification and preservation of the nerve which must be carefully isolated in order to avoid damage to the nerve and to its capsule also as this is often the cause of post-operative pain syndromes (shoulder and neck dysfunction) [20].

Injuries to other nervous structures, such as the cervical plexus and the phrenic nerve, are extremely rare and can be avoided by identifying them and maintaining a superficial position in relation to the deep cervical fascia [21].

Another possible use of IONM is the prevention of marginal mandibular nerve injury during neck dissection in cases where a level I B dissection and thorough clean-up is required [22].

### 3.2. Hypoparathyroidism

Hypoparathyroidism remains the most frequent complication after neck dissection for differentiated thyroid cancer. It can be temporary (23.2–51.3%), but the permanent form can reach important percentages (0.6–15.2%) [23]. The factors contributing to the loss of parathyroid functions include their involuntary removal, vascular damage and thermal energy related damage.

The extension of the surgical dissection up to level VI increases in its incidence [24], as does simultaneous thyroidectomy surgery [25]. The inferior thyroid arteries supply blood to the parathyroid glands via thin and terminal branches that come from the inferior thyroid artery with a contribution, in about 15% of cases, from the branches of the superior thyroid artery that reach the superior parathyroid glands [26]. When the lymphatic structures of the neck central compartment are involved, the lower parathyroid glands can be more easily damaged or removed.

Instead, the superior parathyroid glands, thanks to their more dorsal location, require less manipulation; therefore, they are less likely to be damaged. It is good practice, once the parathyroid glands have been identified, to perform afferent artery ligation near the thyroid capsule, avoiding the use of cautery. In addition, microsurgical magnifying eyeglasses can improve the quality of dissection. In case of impossibility to avoid impairment of the glands or their blood supply, they can be removed and after an intraoperatory histological examination, can be promptly re-implanted in the sternum cleidus mastoid muscle to start functioning in a few months. The performance of neck dissection with a “lateral” approach through the extra-thyroidal space has also been proposed [27].

The development of technologies for the identification and preservation of the parathyroid glands is certainly useful in trying to reduce post-surgical hypoparathyroidism.

Currently, there are two main techniques proposed, namely, near-infrared auto fluorescence [28] and indocyanine green angiography [29]. Both have proved to be useful not only for the identification of the parathyroid glands but also in recognizing and preserving their vascularity. They are relatively simple methods, with a minimum burden in terms of surgical time required and economic expense.

Intraoperative visualization of the parathyroid using near-infrared (NIR) light was first described at Vanderbilt University [30]. The endogenous fluorophores in the parathyroid tissue emit a fluorescent signal when exposed to NIR light. This spontaneous signal from the parathyroid tissue allows the gland to be identified at any time during surgery through an NIR (Fluobeam ^®^ Fluoptics Grenoble France) light display system which is transmitted on a screen. It can also be useful in identifying inadvertently removed glands that need to be reimplanted.

Angiography with indocyanine green (ICG) was initially used for the diagnosis of macular degeneration. Later, it was used for the visualization of lymphatic structures, of the biliary tract and, more recently, to evaluate the vascular blood flow of intestinal anastomoses [31].

An iodinated contrast medium with a half-life of 3–5 min is administered intravenously. It can be monitored through a near-infrared NIR emitter–detector camera system placed above the exposed glands that displays the arterial supply and venous drainage of the parathyroid gland.

These new technologies seem to be able to improve the surgical strategy of recognition and preservation of the parathyroid glands.

### 3.3. Thoracic Duct Injuries

Thoracic duct injury is a serious complication of neck dissection. Fortunately, it is relatively rare (0.6–4.5%); although, when present, it is challenging and difficult to remedy [32]. Generally, the thoracic duct rises 2–3 cm above the clavicle and then descends and enters the deep left jugular vein [33]. Thoracic ducts anatomical variations are reported in Figure 2. In two thirds of the cases, the thoracic duct passes posteriorly to the jugular vein and to the common carotid artery. The anatomical variations mainly concern its outlet into the subclavian vein or into the superficial jugular vein.

Intraoperatively, the abdominal compression maneuver can be useful. In addition to allowing a better visualization of the duct, it also helps to promptly identify and adequately treat a duct injury [34]. Even in the case of the thoracic duct, new technologies can help reduce the incidence of complications. Testimonials have shown that the use of n-Butyl-2-Cyanoacrylate and lymphoscintigraphy with indocyanine green can reduce duct injuries and lymphatic drainage after neck dissection [35].

### 3.4. Vascular Injuries

Vascular injuries during neck dissection for differentiated thyroid cancer are minimal (0.5–1%). Anomalies of the great vessels of the neck are not very common and can be highlighted in the pre-operative instrumental study. The most common is the duplication and fenestration of the deep jugular vein which may be a site of surgical injury [36]. While carrying out the dissection, especially when dealing with level IV, it is important to avoid bending the deep jugular vein. Most vascular injuries occur with the neoplastic invasion of the structures, often in the cases of recurrences that involve levels VI and VII.

## 4. Discussion

Latero-cervical metastases are reported in 30–60% of patients with well-differentiated thyroid cancer [37]. Their presence alone does not seem to have a clinical relevance nor to significantly affect long-term survival [38], this should probably be related to other risk factors, such as the size of the primary tumor, extra-nodal extension and, above all, the age of the patient [39]. In the presence of clinically detectable metastases, it is advisable to perform a therapeutic neck dissection for the local control of the disease. The cervical lymph nodes are divided into groups; historically, their localization divides the neck area into 5 levels, while levels VI and VII, which refer to the central and paratracheal region, have been described more recently [40]. There is no unanimous consensus, however, for what concerns the extent of the dissection. Functional lymphadenectomy is generally indicated in metastases from well-differentiated thyroid cancer, as it allows the sparing of vascular and nerve structures. It generally involves levels I–V, while selective neck dissection includes levels II A, III, IV and V B [41]. These various indications arise from the need to reduce the incidence of post-operative complications while obtaining a good local dissection and clean up from disease. The extension of the lymphadenectomy surely affects the frequency of local recurrences which however do not seem to affect long-term survival [42]. Neck dissection is associated with a high incidence of complications affecting nerves, glands and vessels. In order to reduce such setback, it is important to follow the correct surgical procedures. In our review, we also mentioned technological innovations which can be used to reduce these complications. The patient who must undergo neck dissection requires a thorough and accurate pre-operative study, which must first ascertain the extent of the disease. The use of instrumental methods, such as ultrasound and CT, can be supplemented by Ultrasound-guided Fine-Needle Aspiration Biopsy (US-FNAB) to confirm the positivity of suspicious lymph nodes [43].

The pre-operative study is also important in the follow-up of patients who have already undergone surgery and in which recurrence can reach up to 29% [44]. The nerve structures that can be injured in this type of surgery are many although the lower laryngeal nerve and the spinal accessory nerve are the main ones to be affected. Surgical experience helps to identify and preserve the lower laryngeal nerve.

In recent years, the use of intraoperative neuromonitoring has become increasingly widespread with the aim to reduce the incidence of nerve injuries. This method is used mainly in benign surgery and the results are discordant. Shuwen et al., in their systematic review, report a slight benefit from the use of IONM without reaching statistically significant values [45]. The results reported by the United Kingdom registry of endocrine and thyroid surgery, on the other hand, show a significant reduction in temporary and definitive nerve injuries with the use of IONM [46]. There are no specific reports on the use of IONM during neck dissection, but we can reasonably assume that its use can be beneficial especially in cases of re-operation. IONM appears to be more useful in reducing spinal accessory nerve injuries. The incidence varies according to the extent of the lymphadenectomy, but when it is necessary to dissect levels II b and V a, it can reach 27% [47]. The surgeon’s ability to identify the course of the nerve is combined with the knowledge of the variable relationships of the nerve with the surrounding structures, especially the internal jugular vein [48]. Indeed, an accurate preoperative study using CT and/or MRI is essential to identify the course of the nerve and any vascular anomalies.

IONM is especially useful in the case of subdivision of the branches directed towards the type 2 trapezius muscle structure; that is, before the nerve enters the sternocleidomastoid muscle. It is near the posterior triangle where nerve injury is most frequent, especially if it runs superficially to the vein [49]. A relevant percentage of shoulder syndrome, especially if temporary, does not depend on a direct injury to the nerve, but rather on a damage to its capsule. Therefore, not only is the identification of the nerve mandatory, but also its careful isolation, avoiding injury to the capsule either directly or indirectly through traction or thermal energy.

The other nerves (i.e., hypoglossal nerve, brachial plexus and phrenic nerve) are rarely the site of injuries, and their identification is facilitated by the anatomical respect of the deep cervical fascia. In the rare case of necessity of level 1b dissection, IONM may be useful to avoid injury to the marginal branch of the facial nerve [22]. Transient and definitive hypocalcemia is another frequent complication of neck dissection. Obviously, in the case of dissection of level VI, its occurrence increases considerably [50]. An optimal surgical procedure must be able to identify the glands and preserve their vascularity. In complex scenarios, this is often not possible, especially for the lower parathyroid glands.

Some technological innovations aiming to better identify and preserve the parathyroid glands have been proposed, for instance the near-infrared autofluorescence [51] and the angiography with indocyanine green [52]. First of all, we must specify that no experiences during neck dissection are reported. Once again, technological innovation has divided the experts. Benmiloud et al. report a significant reduction in postoperative hypocalcemia and improved parathyroid identification with the use of near-infrared autofluorescence [53] and Vidal Fortuny emphasizes the usefulness of angiography with indocyanine green in recognizing parathyroid glands with proper function [54].

Kahramangil et al. compare the two techniques and demonstrate an equivalent high percentage of gland identification [55]. Instead, Palazzo et al. believe that currently there are no data that prove, with absolute certainty, the usefulness of near-infrared autofluorescence [28] while Seok Kim reports incidence rates of post-operative hypocalcemia that are not statistically significant after using angiography with indocyanine green [56].

Prospective studies and large numbers are required in order to reach reliable conclusions. Thoracic duct injuries are not frequent but when they occur, they are difficult to remedy. As we know, the thoracic duct must be identified when dissecting level IV lymph nodes on the left side, where it flows into the deep jugular vein after passing behind it. The abdominal compression maneuver can certainly make it more evident as can the use of indocyanine green near-infrared lymphangiography [57]. In order to reduce the incidence of injuries, various measures have been implemented such as the intraoperative application of inactivated Pseudomonas aeruginosa [58] or n-Butyl-2-Cyanoacrylate [35]. These are limited experiences, although they do allow us to envisage possible, more extensive applications. For informational purposes only, experiences that report a different surgical approach for neck dissection through robot, endoscopy or minimal invasive video-assisted thyroidectomy should be recalled [59,60,61]. Obviously, in such reports, the safety and feasibility of these approaches and their undeniable cosmetic advantages are highlighted.

## 5. Conclusions

Lateral neck dissection is recognized as a valid procedure in the local management of differentiated thyroid cancer metastases. This type of surgery is indicated only in cases of clinically detectable metastases given that, in relation to other parameters, it seems to improve the survival rate of these patients. Neck dissection is burdened by a high rate of complications, even in the presence of a proper surgical technique, that can affect glands, vases and nerves. In order to reduce the incidence of these complications, less extensive lymphadenectomies have been recommended, so we outline a functional neck dissection and a selective neck dissection. The downside of this type of lymphadenectomy is the high number of recurrences, which often require re-operation. Therefore, an accurate pre-operative study is necessary in order to accurately identify the extent of the disease and thus plan the most suitable type of lymphadenectomy.

The correct surgical technique involves the knowledge of the relationships between the structures and their possible anatomical variations. Even if the surgical procedure is carried out correctly, the rate of complications still remains high. In recent years, several technological innovations have been proposed to reduce these complications. Neuro-monitoring and parathyroid fluorescence definitely represent a very interesting novelty. These methods are constantly updated and must necessarily be validated with prospective studies based on significant case histories. To conclude, we can state that the possibility of reducing the incidence of complications after neck dissection for differentiated thyroid cancers will certainly rely on a correct surgical technique, and will increasingly make use of new technologies. Moreover, given that there is a lack of literature about this topic, we hope that the present work may give an insight about improvements in surgical technique and technology to help endocrine surgeons in avoiding complications during lateral neck dissection.

## Figures and Tables

**Figure 1 cancers-13-03379-f001:**
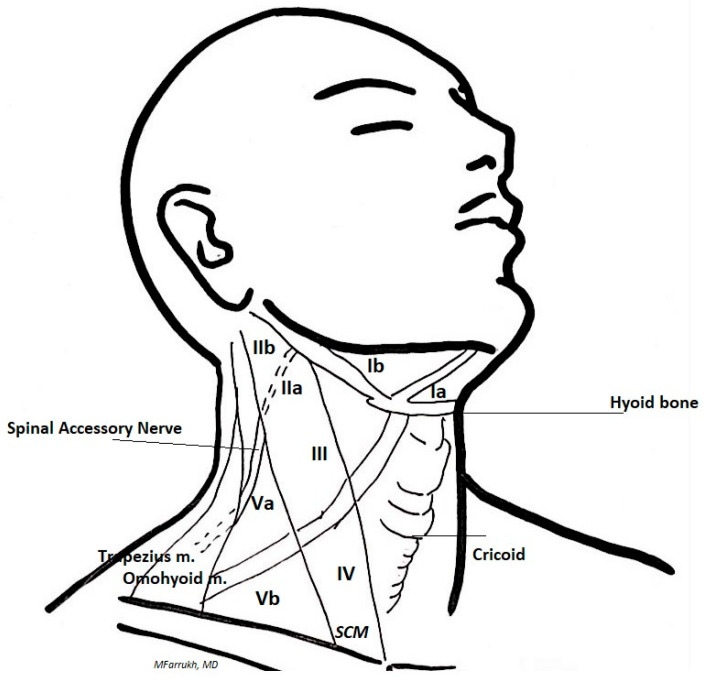
Lymph node levels of the neck and landmarks for lymphadenectomy.

**Figure 2 cancers-13-03379-f002:**
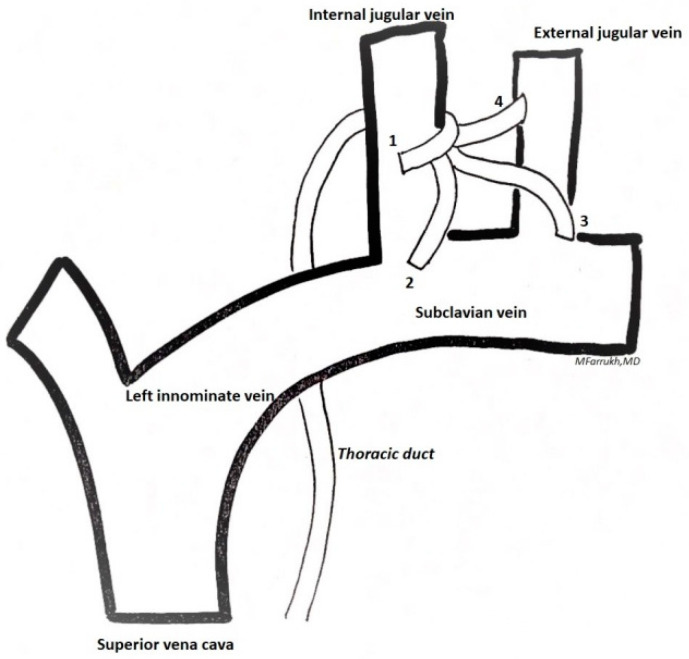
Variations in confluence of the thoracic duct in the venous system: 1—internal jugular vein; 2—jugulovenous angle; 3—subclavian vein; 4—external jugular vein/other.

**Table 1 cancers-13-03379-t001:** Complications of neck dissection for thyroid cancer.

**Central Neck Dissection**
(1) Hypoparathyroidism—temporary/permanent (2) Recurrent laryngeal nerve injury (3) Superior laryngeal nerve injury
**Lateral Neck Dissection**
(1) Hypoparathyroidism—temporary/permanent (2) Chyle leak (3) Hemorrhage (4) Seroma (5) Wound infection (6) Nerve injuries—accessory, ramus mandibularis, sympathetic (Horne’s syndrome), phrenic, brachial plexus, cutaneous cervical plexus

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
