# Peer review of "Surgical Skills and Technological Advancements to Avoid Complications in Lateral Neck Dissection for Differentiated Thyroid Cancer"

_cancers, 2021, doi:10.3390/cancers13143379_

Round 1
Reviewer 1 Report
COMMENTS
The manuscript titled “Surgical skills and technological advancements to avoid complications in lateral neck dissection for differentiated thyroid cancer” of Aldo Bove et al., presents an interesting review on understanding the surgical procedure reserved for thyroid cancer cases with clinically evident lymphatic invasion. Mainly, this study provides a summation of significant morbidity related with neck dissection. Pre-operative plan and surgical experiences comminated with specific technologies are essential to help reduce the percentage of complications. To conserve the integrity of the laryngeal nerve and to preserve parathyroid glands and thoracic duct are skills require to surgeons.
The sections of Introduction and Material and Methods provide sufficient information for the description and reproduction of the study.
However, the following sentence has to be better elaborated: “The initial attempts of the therapeutic removal of the affected lymph nodes date back to 1906 when radical latero-cervical lymphadenectomy was first described in 1906 by George Crile and implied not only the removal of lymph nodes from levels I-V but also the sacrifice of the deep jugular vein, the spinal accessory nerve and the sternocleidomastoid muscle [6]; later, a functional lymphadenectomy was described by the school of Osvaldo Suarez, that allowed the preservation of the above mentioned structures and it is currently the approach used for differentiated thyroid cancer”.
The Results are clearly presented and the Discussion is supported by the results obtained. The Figures also give a helpful visual representation of the case. References are in accordance with guidelines.
However, the following sentence has to be better elaborated: “The surgical technique implies the identification and careful dissection of the nerve, but injury can also be caused indirectly by traction maneuvers or from the use of thermal energy devices”
I have a suggestion to improve the discussion section:
Damage to a recurrent laryngeal nerve can cause problems of voice. Temporary hoarseness, voice tiring, and weakness can occur when one or more of the nerves are irritated during the operation or because of inflammation that occurs after the surgery.
The following references could be consulted:
doi: 10.1136/bcr-2013-201033
doi: 10.21037/gs.2017.06.06
doi: 10.23736/S0392-6621.18.02179-3
In my opinion the manuscript needs minor revision.
Decision:
This study may be accepted for publication.
Author Response
Replies to the Reviewer #1:
According to the reviewer#1 suggestions we have provided to change the following sentences (see comments in the manuscript):
- We think that the reviewer #1 suggestions to improve the “discussion section” are redundant and do not increase the appeal of the aforementioned section.

Reviewer 2 Report
This review article discusses surgical skills and technological advancements in order to help avoid complications in lateral neck dissection for patients with DTC. The article is well written and I have no major comments or concerns, however have a few suggestions for the authors which I believe may strengthen the article:
- In the Introduction the first sentence remove the word "seems".
- What is the difference between the last sentence of the Introduction and Material and Methods section - maybe make this a bit clearer how they fit together.
- Table 1 seems a bit weak - suggest adding a prevalence estimate on how common these are/often these occur from the literature.
- Is there a legend for Figure 1?
- In the Discussion/Conclusion could the authors make it a bit more clear/touch on further how the information in this review may impact on current and future policy and practice?
Author Response
Replies to the Reviewer #2:
According to the reviewer#2 suggestions we have provided to change the following sentences (see comments in the manuscript):
- We have provided to change the word “seems” with the word “is”.
- We have modified this part according to the reviewer #2, moving the sentence from the introducion section to the materials and methods one. We hope that this change could better explain the concept.
- The incidence of the complications are clearly reported in the manuscript. The choice to not report the percentages in the table has stylistic reasons, that is not to make the contents of the table itself redundant.
- According to the reviewer #2 suggestion, we have provided to make the legend of the Figure 1 more understandable (see in the comment)
- According to the reviewer #2 suggestion, we have provided to add this sentence to the conclusion section to better explain how this paper could impact also in experienced endocrine surgeons clinical practice.